# Influence of Dietary Fibre and Protein Fractions on the Trace Element Bioaccessibility of Turnip Tops (*Brassica rapa*) Growing under Mediterranean Conditions

**DOI:** 10.3390/foods13030462

**Published:** 2024-02-01

**Authors:** Fernando Cámara-Martos

**Affiliations:** Departamento de Bromatología y Tecnología de Alimentos, Universidad de Córdoba, 14014 Cordoba, Spain; fernando.camara@uco.es; Tel.: +34-957-212026

**Keywords:** trace elements, bioaccessibility, pectin, gum arabic, cellulose, casein, lactalbumin, soy protein

## Abstract

The objective of this work was to study the influence of three dietary fibre fractions (pectin, gum arabic and cellulose) and three protein fractions (casein, lactalbumin and soy) on the trace element bioaccessibility (Fe, Mn, Ni, Se and Zn) of turnip tops (*B. rapa* subsp. *Rapa*) growing under Mediterranean conditions. Then, it aimed to promote the use of this vegetable not only for direct fresh consumption but also as a main ingredient in the development of food mixtures. The results showed that soluble fibre fractions, such as pectin and gum arabic, can enhance the bioaccessibility of trace elements, such as Fe, Mn, Se and Zn. This effect was not proved for cellulose (an insoluble fibre fraction), in which, at best, no bioaccessibility effect was observed. Regarding the protein fractions, with the exception of Se, caseins and lactalbumin had a neutral effect on improving the trace element bioaccessibility. This did not hold true for soy protein, in which a considerable improvement in the bioaccessibility of Fe, Mn, Se and Zn was determined.

## 1. Introduction

In recent years, diets common amongst populations of Western countries have been based on a high consumption of meat, which is to the detriment of fruit, vegetable and legume consumption. Numerous studies have shown that this excessive ingestion of meat, particularly red meat, is a risk factor in the development of cancers, cardiovascular diseases, obesity and type 2 diabetes mellitus [1,2,3,4]. In addition, for religious, cultural and ideological reasons, the adoption of a vegetarian or vegan diet is becoming increasingly popular [5]. Furthermore, the growing world population and the ecological impact of meat production in terms of greenhouse gas emissions, energy, water and land use are also significant. Thus, when considering the aforementioned, health authorities should encourage greater consumption of vegetables. Moreover, the food industry can assist by developing new products based on vegetable food mixtures.

Vegetable species belonging to the *Brassicaceae* (formerly *Cruciferae*) family are some of the most economically important plant groups for humans, with varieties such as broccoli, cabbage, cauliflower, rapeseed, mustard and rocket [6]. From a nutritional point of view, they are unique as they contain a group of phytochemicals called glucosinolates which, when they are hydrolysed in the digestive tract, produce, amongst other compounds, isothiocyanates. These compounds stimulate anticarcinogenic phase II human enzyme activities [7,8,9]. Furthermore, previous studies [10,11,12] have shown that unlike other green leafy vegetables (such as spinach or Swiss chard), the trace element bioaccessibility present in cruciferous vegetables is notably high, similar in some cases to that of powdered milk. This is due to a low content of anti-nutritional compounds such as oxalates.

*Brassica rapa* L. is another species belonging to the *Brassicaceae* family. Its cultivation covers a wide geographical area ranging from Central Asia to the western Mediterranean region [13,14]. In northwest Spain and Portugal, there has been a long tradition of cultivating *B. rapa* subsp. *rapa* to obtain turnip tops. The latter are fructiferous stems with flower buds and surrounding leaves that are consumed whilst still green [15]. This vegetable is widely consumed in this part of the Iberian Peninsula and is even the main ingredient in traditional recipes. However, in the southern region of the Mediterranean basin, its cultivation has been very limited, which is likely due to its lack of adaptation [16]. In spite of this, in recent years, a programme for adapting varieties of *B. rapa* subsp. *rapa* to the climatic and environmental conditions of southern Spain has been developed with considerable success [16,17].

The next step is the use of this vegetable not only for direct fresh consumption but also as a main ingredient in the development of food mixtures. To this end, it would be of great interest to study how the presence of other ingredients could affect the trace element bioaccessibility present in turnip tops. Bioaccessibility refers to the fraction of micronutrient or bioactive compound (bioaccessible fraction) released from the food matrix and solubilised in the intestinal lumen during the digestive process, thereby becoming potentially available for absorption in the small intestine [18,19,20]. Bioaccessibility can be determined from simulated in vitro gastrointestinal digestion by which the conditions in the human digestive tract are reproduced. In recent years, an attempt has been made to achieve a certain standardisation across bioaccessibility methodologies [21,22]. However, the latter may overestimate trace element bioaccessibility due to the physiological fluids used [23].

As already stated, trace element bioaccessibility could be influenced by the presence of other dietary components such as proteins, fat, dietary fibre or even other inorganic elements [24,25,26]. Therefore, the aim of the present work was to study the influence of three dietary fibre fractions (pectin, gum arabic and cellulose) and three protein fractions (casein, lactalbumin and soy) on the trace element bioaccessibility of turnip tops (*B. rapa* subsp. *rapa*) growing under Mediterranean conditions. To our knowledge, extensive studies on the influence of protein and fibre fractions on the trace element bioaccessibility are scarce. The results of this research could be taken into account when formulating food mixtures in order to preserve the excellent nutritional properties of these vegetables.

## 2. Material and Methods

### 2.1. Material and Reagents

All reagents used were of analytical grade such as hyperpure HNO_3_ (69%) (Panreac, Barcelona, Spain). All traces of metallic elements on the plastic and glass material were decontaminated by immersing them in a 50% HNO_3_ bath for 24 h. This material was then transferred to and left in a 20% HCl bath for a further 24 h. Finally, the plastic and glass material was rinsed three times with deionised water and allowed to dry. Finally, all of the experiments were carried out with deionised water (resistivity 18 µS/cm) obtained using a Milli Q Reference Water Purification system (Millipore, Madrid, Spain).

Sodium bicarbonate (97%) was obtained from Scharlau (Barcelona, Spain), magnesium nitrate hexahydrate (98%) and magnesium oxide (98%) from Alfa Aesar (Kandel, Germany). Hydrochloric acid (35%) and hyperpure nitric acid (65%) were obtained from Panreac (Barcelona, Spain). Sodium borohydride, digestive enzymes and bile salts were supplied by Sigma-Aldrich Co. (St. Louis, MO, USA) as follows: α–amylase from human saliva type IX–A, 1000–3000 U/mg (Ref. A0521); pepsin from porcine gastric mucosa ≥ 250 U/g (Ref. P7000); pancreatin from porcine pancreas ≥ 100 U/g (Ref. P3292); and bile salts (Ref. B8756). Working solutions of these enzymes were prepared immediately before use. Pectin, gum arabic, cellulose, casein, lactalbumin and soy protein were supplied by Sigma-Aldrich Co. (St. Louis, MO, USA).

### 2.2. Samples

Samples of turnip tops were either purchased or provided by local farmers. They were growing in a typically continental Mediterranean climate (Csa in Köppen’s classification) which is characterised by relatively cold winters, intensely hot dry summers, and a mean annual precipitations of 650 mm. Upon arrival at the laboratory, the samples were washed with deionised water to remove any traces of soil and dirt. Subsequently, the turnip tops were placed in paper bags, frozen, and lyophilised at −50 °C using a Scanvac Labogene, model COOLSAFE 55- 4 BASIC for 10 days. Finally, the lyophilised turnip tops were ground into a fine powder using a ceramic ball mill (OABM 255 model; Orto Alresa; Madrid, Spain).

### 2.3. Bioaccessibility Assay

The bioaccessibility determination (soluble fraction) was carried out using the procedure described by Cámara et al. [27] with slight modifications. This procedure replicates the physiological conditions occurring in the human digestive tract, including enzymes, pH, temperature and the duration of digested food in the mouth (oral phase), stomach (gastric phase) and intestine (intestinal phase). For this, 2 g of turnip tops were homogenised with 23 mL of deionised water (control assay) in a glass bottle (150 mL). For the assays conducted with different fibre fractions (pectin, gum arabic, cellulose) and protein fractions (casein, lactalbumin, soy), alongside the 2 g of turnip tops, increasing amounts of the aforementioned compounds were added to represent 5, 15 and 25% in the final mixture, i.e., 0.105, 0.353 and 0.666 g, respectively.

Subsequently, 500 µL of α-amylase solution (1500 U/mL) was added. The mixture was placed in a thermostatic bath with agitation (HSB-2000 Shaking Bath; E-Chrom Tech CO., ltd, Taipei, Taiwan) at 37 °C for 2 min (oral phase). After this time, the pH was lowered to 2 with 6N HCl and 0.083 g of pepsin solution in 0.52 mL HCl 0.1 N was added to each sample. The sample was then reincubated for 2 h at 37 °C in a thermostatic bath with agitation (gastric phase).

For the simulation of the intestinal stage, the pH was adjusted to 5 by adding 1M NaHCO_3_. Then, 4.2 mL of a mixture of pancreatin and bile salts (equivalent to 0.017 g of pancreatin and 0.107 g of bile salts in 4.2 mL NaHCO_3_ 0.1N) were added to each sample. The digested material was once again incubated in a thermostatic bath with agitation at 37 °C for an additional 2 h. Finally, the pH was adjusted to 7.2 with 0.5 M NaOH. The final mixture volume was around 40 mL The digested material was transferred to 50 mL polypropylene centrifuge tubes and centrifuged (Eppendorf Centrifuge 5810 R) at 4000 rpm for 1 h at 3 °C. Finally, the supernatant was collected to determine the concentration of trace elements present in it (bioaccessible fraction).

All the assays were made by triplicate. In order to eliminate the bioaccessible trace element fraction contributed by the fibre and protein fractions in the final mixture, all assays were conducted with blanks for each of the protein and fibre fractions, as well as for each proportion. That is, bioaccessibility test with 0, 0.105, 0.353 and 0.666 g of each of the protein and fibre fractions (without turnip tops) were also carried out.

### 2.4. Trace Element Analysis

For the determination of trace elements, the bioaccessible fraction obtained in the previous procedure was placed in porcelain crucibles and evaporated to dryness on a hot plate at 80 °C. To prevent Se volatilisation, samples were treated with 5 mL of 7 M HNO_3_ and 1.5 mL of ashing aid suspension (20% *w*/*v* MgNO_3_ and 2% *w*/*v* MgO). The residue was incinerated in a muffle furnace at 450 °C for 16 h. The resulting ashes were bleached by adding 200 µL of hyperpure HNO_3_ and 2 mL of deionised water, heating to dryness on a hot plate at 80 °C and reintroducing them into the muffle furnace at 450 °C for an additional 1 h. Ashes were recovered with 100 μL of hyperpure HNO_3_ made up to 10 mL with deionised water. For the determination of the total content of trace elements present in turnip tops, a procedure similar to that carried out with the bioaccessible fraction was conducted using 0.5 g of the sample.

Fe (λ = 248.3 nm) and Zn (λ = 213.9 nm) were determined by flame absorption atomic spectroscopy (FAAS) with a Varian Spectra AA–50B model (Palo Alto, CA, USA) equipped with single element hollow cathode lamps and a standard air-acetylene flame. Mn (λ = 403.1 nm) and Ni (λ = 232.0 nm) were determined by electrothermal absorption atomic spectroscopy (ET–AAS) (Agilent Technologies model 240Z AA; Santa Clara, CA, USA) with a graphite furnace and autosampler. The instrumental conditions for the analysis of these elements were similar to those reported in previous studies [10,13,25,26].

Finally, the Se content of the samples was determined by atomic fluorescence spectroscopy (AFS) (λ = 196.0 nm) (Millennium Excalibur Instrument, PSA Analytical, Kent, United Kingdom). The formation of Se hydride was carried out by pumping 0.7% *w*/*v* NaBH_4_ (in 0.1 M NaOH) and 4.5 M HCl with the flow rate set at 10 mL/min. A gas–liquid separator and Argon gas (300 mL/min) was used to transport the selenium hydride.

Standard solutions for quantifying Fe, Mn, Ni, Se and Zn were prepared prior to usage. These solutions were created by diluting with deionised water 1000 mg/L standard solutions obtained from Certipur, Merck (Darmstadt, Germany). The samples were analysed in duplicate. Calibration curves were obtained from five concentration levels between 0.25 and 4.0 mg/L for Fe and Zn; 10 and 100 µg/L for Mn and Ni; and 0.05 and 1 µg/L for Se. The accuracy and precision of the different analytical techniques used in determining trace element concentrations was validated by recovery experiments using Certified Reference Materials (see Table 1). Recovery percentages ranged between 90 and 109% for all the studied elements. Limit of detection (LOD): Fe (0.03 mg/L); Mn (0.88 µg/L); Ni (0.79 µg/L); Zn (0.03 mg/L). Limit of quantitation (LOQ): Fe (0.09 mg/L); Mn (2.92 µg/L); Ni (2.63 µg/L); Zn (0.12 mg/L).

### 2.5. Statistical Analysis 

The IBM SPSS 25 statistical software package was used for statistical analysis. The data were expressed as mean and standard deviation. Data were analysed using ANOVA tests. Significant differences were considered when *p* < 0.05.

## 3. Results and Discussion

The total trace element content in turnip tops was 88 mg/Kg for Fe; 44 mg/Kg for Mn; 2.55 mg/Kg for Ni; 53 µg/Kg for Se; and 47 mg/Kg for Zn. These results are in agreement with those obtained in previous studies [10,15,28,29]. The contents of bioaccessible trace elements in the assay for turnip tops alone, without adding any other components (control) was 28.4 mg/Kg for Fe; 21,9 mg/Kg for Mn; 1.05 mg/Kg for Ni; 9.2 µg/Kg for Se; and 16.7 mg/Kg for Zn.

As indicated in the Material and Methods section, all the results obtained below were performed using blanks (reagents + protein or fibre fraction). The aim was to eliminate the bioaccessible trace element fraction contributed by the reagents and the added fractions (for each of the percentages added).

### 3.1. Pectin

Pectin increased the trace element bioaccessibility for Fe and Se from the 5% dose. In the case of Zn, this increase was observed from the 15% dose and for Mn, from the 25% dose (see Table 2). In general, pectin improved the bioaccessibility of these nutritionally relevant trace elements. Pectin is a polymer of methyl D-galacturonic acid, linked by glycosidic (ß 1–4) bonds. The main chain also has segments of L-rhamnose linked to the galacturonate and in smaller amounts D-galactans and L-arabinanes [30,31]. In addition, the carboxylic groups of pectins are in a methyl ester form in different proportions (methylesterified galacturonic acid residues). Others can also be in the free-acid form (as non-methylesterified galacturonic acid residues). This causes pectin products to be subdivided according to their degree of methylation, which is the percentage of carboxyl groups esterified with methanol [30,31].

It has been proposed that the percentage of methylesterified residues is related to the ability of pectin to uptake minerals and trace elements [32,33]. When the pH is above the pKa of pectin (3.38–4.10), most of the galacturonic acid residues are found as non-methylesterified ones, and the capacity to uptake trace elements cations increases [33]. Accordingly, at the pH of 5.5, under which the intestinal stage of our bioaccessibility assay was performed, the ionisation of the carboxyl groups is high and so is their capacity to uptake trace elements cations. Similar behaviour could be expected for those bioaccessibility protocols that use a pH 7 for the intestinal stage. In addition to the non-methyl-esterified galacturonic acid residues, the ion uptake properties of pectin have also been linked to other functional groups such as the free hydroxyl ones [34].

The results obtained in this work could justify that this uptake of trace elements by pectin, one of the soluble fractions of dietary fibre, would have a positive effect on its bioaccessibility. The trace elements bound by the solubilised pectin would also remain soluble in the intestinal lumen preventing them from precipitating as non bioaccessible insoluble salts. This effect has also been observed in a previous study by Peixoto et al. [35]. In the latter, the presence of pectin improved the bioaccessibility of trace elements (Fe, Zn and Mn among others) in a chocolate drink powder. According to these authors, the presence of some anti-nutritional factors, such as phytates, could decrease the absorption of some essential elements due to the formation of precipitates. The formation of soluble complexes between these elements and pectin, which is a soluble fibre, prevents such precipitation [35]. In other words, pectin would keep the trace elements solubilised until they are taken up by enterocytes. In addition, Miyada et al. [36] has suggested that under physiological conditions in the small intestine, pectin has the ability to reduce ferric Fe to ferrous form. Within the two forms of non-heme Fe, ferrous Fe is absorbed more efficiently than ferric Fe due to its greater solubility. These results are not in agreement with those reported in other studies that have attributed pectin with an inhibitory effect on the trace element bioaccessibility that is present in foods such as beans [37,38]. The different food matrix in which the effect of pectin was studied may be responsible for these differences.

Ni was the only element studied for which pectin did not improve its bioaccessibility. This result is even considered to be a positive one as Ni is considered to be a heavy metal rather than an essential trace element [28]. Some authors have indicated that the affinity of a trace element for pectin depends on its electronegativity [39]. Although this would not justify the differences found for Ni. In this respect, it should be noted that the formation of soluble or insoluble complexes in the gastrointestinal tract is conditioned by numerous factors such as pH, oxidation state of the element and the presence or absence of other trace elements [35].

### 3.2. Gum Arabic

The effect the presence of gum arabic has on trace element bioaccessibility was similar to that of pectin (see Table 3). With the exception of Fe, gum arabic improved the bioaccessibility of Mn, Se and Zn from the 5% dose. For Fe, an increase in its bioaccessibility was also observed, although the differences were not statistically significant. Furthermore, as with pectin, Ni bioaccessibility decreased with the dose of gum arabic added.

Gum arabic is widely used as an ingredient in the pharmaceutical and food industry [40]. It is a complex polysaccharide consisting mainly of galactopyranose and arabinopyranose, with 2% protein as an integral part of its structure. It is precisely this protein content that gives rise to three gum arabic fractions: a majority fraction (90%) with a very low protein content; a second fraction with a protein content of 10% (10% of the total); and finally, a minority fraction (below 1%) with a protein content of 50% [41]. It is precisely the second fraction that is used as an emulsion stabiliser due to its affinity to adsorb at the oil–water or air–water interface. In this regard, gum arabic has been used as a natural functional ingredient for the fortification of trace elements such as Fe and Zn [42], wherein these ions are chelated by the self-assembled polymer host. Similarly, Li et al. [43] have shown that gum arabic facilitates the endocytic uptake of ferric oxyhydroxide nanoparticles by micropinocytosis. These authors, in an animal model (rats), also found a higher Fe bioavailability and serum Fe concentration for a gum arabic–ferric nanoparticle mixture versus ferrous sulphate. The findings of these previous works agree with those of our study in that the presence of gum arabic improved trace element bioaccessibility.

### 3.3. Cellulose

Cellulose is the most abundant compound in plant cell walls. In vegetables, its values range between 0.2–5% of raw product [44]. In cereals, it is found in the pericarp in a percentage of around 30% [45]. The influence of cellulose on micronutrient bioaccessibility was different according to the trace elements studied. For Mn, Se and Zn no effect was observed whereas for Fe and Ni a decrease in the bioaccessibility of these elements was observed as from the 5% dose (Table 4). Unlike the two fractions previously studied (soluble fibre fractions), cellulose is categorised as the insoluble fraction of dietary fibre. The ability of the fibre matrix to adsorb metal ions depends on their types and characteristics and those of the metal ion [46,47,48]. 

It has been pointed out that the ability of insoluble dietary fibre to adsorb metal ions is lesser than that of soluble fibre [49], even sometimes negligible [50]. This could be the reason why no effect on the Mn, Se and Zn bioaccessibility was observed as these trace elements were not retained by the cellulose. On this point, a recent study [51] has found that microcrystalline cellulose (similar to that used in this study) had no effect on the bioaccessibility of Ca^+2^ and Zn^+2^ ions. The same study [51] also found that microcrystalline cellulose had no effect on serum Ca^+2^ and Zn^+2^ levels in rats nor on their bone density and strength, compared to the control group.

However, for Fe and to some extent Ni, a decrease in their bioaccessibility was observed as the cellulose dose increased. As already mentioned, the capacity of fibre to uptake or adsorb metal ions depends on the characteristics of the fibre and the metal ion. In contrast to microcrystalline cellulose, the study by He et al. [51] observed that carboxymethyl cellulose and other cellulose modified forms impaired Ca^+2^ and Zn^+2^ ions bioaccessibility compared to a control group. Similarly, Plait et al. [52] have indicated that both cellulose and other forms of insoluble fibre (lignin) have a high capacity to bind Fe in an insoluble form at a pH between 5.7–6.6. Indeed, this Fe chelating ability has also been associated with other insoluble fibre fractions. Thus, Horniblow et al. [53] have recently shown in cell line models (RKO and Caco–2 cells) a significant decrease in intracellular Fe concentration and ferritin expression in the presence of lignin. In our study, we hypothesised that adsorption of Fe to insoluble cellulose also led to insolubilisation of this element in the gastrointestinal model proposed, thus decreasing Fe bioaccessibility.

### 3.4. Casein

The influence of casein on the bioaccessibility of Fe and Mn in turnip tops was negligible. In the case of Zn, the improvement in its bioaccessibility was only effective as from the highest dose (25%) (see Table 5).

Casein is a high-quality protein that is present in dairy products at different fractions (alpha, beta and kappa) [54,55]. It can represent around 83% of the protein fraction in goat’s or cow’s milk, whereas serum proteins represent 17% [56]. The role of caseins in trace element bioaccessibility remains unclear, with some authors indicating that these proteins may have an inhibitory effect on Fe absorption [57]. Nevertheless, a new Fe fortifier has recently been developed by combining sodium caseinate and ferric chloride in the presence of orthophosphate [58,59]. The intake of this Fe-casein complex with whole milk has proved in healthy young women to have an Fe bioavailability, in terms of serum ferritin, that is similar to ferrous sulphate [60]. Similarly, Sabatier et al. [61] have shown a higher Fe uptake by cell lines (Caco–2 cells) from aqueous solutions of that Fe-casein complex than from ferrous sulphate. The latter is a traditional Fe fortifier but can induce undesirable organoleptic changes. Furthermore, in an in vitro digestion model, it has been observed that when the pH increases during the intestinal stage, ferrous sulphate can insolubilise and become non-absorbable Fe hydroxide [62]. On the contrary, the Fe-casein complex is a colloidal one that prevents the precipitation of Fe and caseinate in aqueous systems [58,59].

On the other hand, in vitro digestion of caseins releases bioactive peptides known as caseinophosphopeptides which may also play an important role in trace element bioavailability [63]. These peptides have a cluster of three phosphorylated serine residues followed by two glutamic acid residues that are potential binding sites for Fe [64]. Other studies have shown that aspartic and glutamic acid, glutamine, phenylalanine, proline, serine, tyrosine and lysine residues may also have important roles as Fe-chelating peptides [65,66,67]. Miao et al. [68] have obtained a peptide from the enzymatic hydrolysis of casein with a high capacity to bind to Fe. For that reason, these authors report that these peptides could be used as functional ingredients to prevent Fe deficiency. Fe uptake by these caseinophosphopeptides would possibly improve the bioaccessibility of this element. However, studies on the effect of caseinophosphopeptides on Fe bioaccessibility yield controversial results, depending on the type of caseinophosphopeptide and the degree of caseins hydrolysis. Thus, peptides derived from β-caseins are more effective for Fe bioavailability than those derived from α-caseins [64]. Furthermore, García-Nebot et al. [64] in cell lines (HuH7 cells), have observed that peptides derived from α- and β-caseins have no effect on Fe bioaccessibility in terms of ferritin and soluble transferrin receptor contents. Our study’s results agreement with those of this latter work. Caseins had a neutral effect (neither positive nor negative) on the Fe bioaccessibility present in turnip tops.

Regarding Zn, an increase in the bioaccessibility of this element was only observed for the casein dose of 25%, whereas at lower doses no effect was observed. Caseins have a natural ability to bind to Zn [55,69,70]. At a neutral pH, the phosphoric and carboxylic acid residues of α-casein and the threonine residues of k-casein have a high affinity for binding to Zn ions [69]. Nevertheless, studies on the effects of caseins on Zn bioaccessibility are contradictory and scarcely studied. Lonnerdal et al. [71] stated that Zn taken up into human milk caseins could improve their bioaccessibility by being delivered as such to intestinal receptors. However, in rats, the same authors have subsequently observed that Zn uptake into bovine milk caseins is the cause of a lower bioavailability when compared to an infant whey formula or human milk [72]. Thus, peptides derived from the digestion of bovine caseins would form complexes with Zn making it unavailable for absorption. In contrast, Peres et al. [73] have observed that the presence in rats of a peptide derived from β casein improves Zn absorption (higher and faster) than that of zinc sulphate. This increase in Zn absorption could be due to the Zn bound to the soluble β-caseinophosphopeptide being protected from insolubilisation during the digestive process or from the negative interaction of other anti-nutritional compounds such as phytates [73].

Caseins had no effect on Mn bioaccessibility either, which is somewhat justifiable considering the low Mn content in milk. On the other hand, it was highly obvious that these proteins caused an important increase in the Se bioaccessibility. This was noted to be from 18% (control) to 96% and from the lowest dose (5%). To our knowledge, no studies have investigated the influence of caseins on Se bioaccessibility. What is known is that Se in milk and dairy products is strongly associated with both caseins and whey proteins [74,75]. In addition, the presence of protein ingredients has also been seen to increase the Se content in food mixtures [76] and Se could be absorbed better from a high protein diet or when protein isolates are presented [77,78]. Based on this, these previous findings could justify the promoter effect of caseins on the Se bioaccessibility present in the turnip tops of our study.

### 3.5. Lactalbumin

The effect of lactalbumin on the trace element bioaccessibility present in turnip tops was very similar to that found for casein. There was none for Fe, Mn and Zn, a slight decrease for Ni and a remarkable improvement in Se bioaccessibility (reaching 85%) from the lowest protein dose (5%) (see Table 6). Lactalbumin is the second most abundant protein in whey, accounting for approximately 2% of milk proteins and 13% of whey proteins. It consists of 123 amino acids and has a molecular weight of 14,146 [79]. Lactalbumin has a single strong Ca-binding site but can also bind to other trace elements such as Cu, Mn or Zn (with several distinct Zn^+2^ binding sites), and even to heavy metals (Pb and Hg) [80]. This specific Zn-binding site could be the reason for a higher Zn absorption and plasma Zn concentration in infant rhesus monkeys fed with an infant formula supplemented with lactalbumin [81]. However, this result was not found in the in vitro assay performed in our study and further clarification is needed in humans.

In agreement with our results, lactalbumin did not affect Fe absorption from any infant formula in healthy term, 6-month-old children [82]. Similarly, lactalbumin hydrolysate did not improve Fe absorption and the ferritin content in a Caco–2 cell line model [83]. Regarding Se, the explanation for the considerable increase in its bioaccessibility upon the presence of lactalbumin could be similar to that provided above for caseins. Protein ingredients are the main dietary source of Se and also could promote its bioavailability [74,75,76].

### 3.6. Soy Protein

Unlike the two previous protein fractions, soy protein improved the bioaccessibility of most of the trace elements studied (with the exception of Ni for which no effect was observed). The bioaccessibility percentages increased from 32% to 93% for Fe; from 50% to 88% for Mn; from 17% to 100% for Se; and from 36% to 100% for Zn (Table 7). These results are in accordance with those reported recently by Milani et al. [84], who showed that soy-based beverages from different soy sources (isolate protein, hydrosoluble extract and beans) could be a significant Fe, Se and Zn dietary source. Similarly, for soybean, Iaquinta et al. [5] have reported high Fe and Zn bioaccessibility with similar percentages to those obtained from a selection of meat samples analysed (beef and chicken).

Soybeans are a favourable source of protein, containing 14 g per 100 g. This content can increase up to 66–70% in soy concentrates [85]. Compared to animal protein, soy is the most widely consumed vegetable protein because of its cardioprotective effect, due to its absence of cholesterol and saturated fat and the presence of polyunsaturated fatty acids and antioxidants [86]. Despite this, the effects of this protein fraction on the trace element bioaccessibility are contradictory., Herrera-Agudelo et al. [87] found low bioaccessibility values for Mn and especially for Fe and Zn in soybean seeds. This low bioaccessibility is attributed to the presence of anti-nutritional factors such as phytates, oxalates, tannins and polyphenols. These compounds would form insoluble complexes with proteins and trace elements (such as Fe and Zn) yielding insoluble complexes at the duodenal pH, hampering its absorption [88,89]. The presence of phytic acid was also the cause of the lower Zn bioaccessibility reported in soy-based beverages (18–21%) versus bovine milk (59–81%) [90]. Phytate hydrolysis could be an interesting process to increase the trace element bioaccessibility in these beverages. Thus, Theodoropoulos et al. [91] observed that phytase treatment significantly improved the bioaccessibility of Fe (from 2.2 to 37.1%) and of Zn (from 38.8 to 67.4%) in a soy drink.

Our results show a promoter effect of soy protein on the bioaccessibility of Fe, Mn, Se and Zn, reaching in some cases around 100%. The differences from the aforementioned studies could be due to our study only using purified soy protein in which the anti-nutritional factors of soybean are not present. Thus, the extraction of this protein from its natural sources and its use as a functional ingredient could be an alternative for developing food mixtures in which the bioaccessibility of trace elements is not impeded. However, more research needs to be done in this direction.

## 4. Conclusions

The soluble fibre fractions, such as pectin and gum arabic, can enhance the trace element bioaccessibility for Fe, Mn, Se and Zn found in cruciferous vegetables such as turnip tops. One of the reasons for justifying this increase could be the capacity of this type of fibre to bind these trace elements and keep them soluble in the intestinal lumen until the moment of absorption by enterocytes. This effect is not applicable to insoluble fibre fractions such as cellulose, in which, at best, no effect is observed. In fact, there might even be a decrease in the bioaccessibility of other trace elements such as Fe. On the other hand, with the highly obvious exception of Se, milk proteins such as caseins and lactalbumin have a neutral effect on improving trace element bioaccessibility.

The above does not hold true for soy protein, as a considerable improvement in the bioaccessibility of Fe, Mn, Se and Zn was observed. The results obtained here could be taken into consideration when selecting various ingredients for formulating new foods developed from cruciferous vegetables.

## Figures and Tables

**Table 1 foods-13-00462-t001:** Analysis of certified references materials (mean ± standard deviation).

Element	Certified References Material (mg kg^−1^)								
	Bovine Liver ERM-BB185			White Cabbage BCR-679			Peach Leaves NIST-1547		
	Certified	Found	Recovery (%)	Certified	Found	Recovery (%)	Certified	Found	Recovery (%)
Fe				55.0 ± 2.5	59.1 ± 1.1	107	219.8 ± 6.8	197.0 ± 0.2	90
Mn				13.3 ± 0.5	13.2 ± 0.8	99	97.8 ± 1.8	92.1 ± 2.3	94
Ni				27.0 ± 0.8	29.6 ± 0.2	109	0.689 ± 0.095	0.681 ± 0.101	99
Se	2.99 ± 0.18	2.69 ± 0.40	90						
Zn				79.7 ± 2.7	79.7 ± 0.2	100	17.97 ± 0.53	16.90 ± 0.46	94

**Table 2 foods-13-00462-t002:** Trace element bioaccessibility (mg/Kg dry weight) in turnip tops with pectin (mean ± standard deviation).

	Fe (mg/Kg)	Mn (mg/Kg)	Ni (mg/Kg)	Se (µg/Kg)	Zn (mg/Kg)
Turnip tops	28.4 ± 6.2 ^a^	21.9 ± 3.9 ^a^	1.05 ± 0.07 ^b^	9.2 ± 2.4 ^a^	16.7 ± 0.7 ^a^
Turnip tops + 5% pectin	44.0 ± 2.9 ^b^	21.0 ± 1.4 ^a^	0.87 ± 0.18 ^b^	21.3 ± 4.6 ^b^	19.1 ± 4.0 ^a^
Turnip tops + 15% pectin	46.5 ± 2.7 ^b^	21.3 ± 0.3 ^a^	0.78 ± 0.08 ^b^	16.4 ± 3.4 ^b^	24.2 ± 6.4 ^a,b^
Turnip tops + 25% pectin	47.5 ± 1.3 ^b^	26.5 ± 3.1 ^b^	0.11 ± 0.03 ^a^	38.0 ± 2.9 ^c^	35.4 ± 9.4 ^b^

Within each column, means with different lowercase letters are significantly different at *p* < 0.05 according to the analysis of variance (ANOVA).

**Table 3 foods-13-00462-t003:** Trace element bioaccessibility (mg/Kg dry weight) in turnip tops with gum arabic (mean ± standard deviation).

	Fe (mg/Kg)	Mn (mg/Kg)	Ni (mg/Kg)	Se (µg/Kg)	Zn (mg/Kg)
Turnip tops	28.4 ± 6.2 ^a^	21.9 ± 3.9 ^a^	1.05 ± 0.07 ^b^	9.2 ± 2.4 ^a^	16.7 ± 0.7 ^a^
Turnip tops + 5% gum arabic	34.0 ± 2.9 ^a^	26.1 ± 2.2 ^a,b^	0.38 ± 0.06 ^a^	21.4 ± 4.0 ^a,b^	23.9 ± 1.2 ^b^
Turnip tops + 15% gum arabic	33.7 ± 3.2 ^a^	27.6 ± 1.1 ^b^	0.18 ± 0.05 ^a^	34.6 ± 11.5 ^c^	26.6 ± 0.8 ^c^
Turnip tops + 25% gum arabic	34.5 ± 1.6 ^a^	29.1 ± 0.9 ^b^	0.20 ± 0.06 ^a^	25.7 ± 5.5 ^b,c^	25.5 ± 0.3 ^c^

Within each column means with different lowercase letters are significantly different at *p* < 0.05 according to the analysis of variance (ANOVA).

**Table 4 foods-13-00462-t004:** Trace element bioaccessibility (mg/Kg dry weight) in turnip tops with cellulose (mean ± standard deviation).

	Fe (mg/Kg)	Mn (mg/Kg)	Ni (mg/Kg)	Se (µg/Kg)	Zn (mg/Kg)
Turnip tops	28.4 ± 6.2 ^b^	21.9 ± 3.9 ^a^	1.05 ± 0.07 ^b^	9.2 ± 2.4 ^a^	16.7 ± 0.7 ^a^
Turnip tops + 5% cellulose	15.3 ± 9.2 ^a,b^	19.0 ± 0.5 ^a^	0.32 ± 0.08 ^a^	10.2 ± 3.4 ^a^	17.7 ± 1.3 ^a^
Turnip tops + 15% cellulose	11.2 ± 1.3 ^a^	18.3 ± 1.8 ^a^	0.35 ± 0.14 ^a^	9.7 ± 1.2 ^a^	16.4 ± 1.1 ^a^
Turnip tops + 25% cellulose	8.1 ± 0.5 ^a^	17.0 ± 0.2 ^a^	0.63 ± 0.06 ^a^	9.8 ± 2.0 ^a^	16.1 ± 0.4 ^a^

Within each column, means with different lowercase letters are significantly different at *p* < 0.05 according to the analysis of variance (ANOVA).

**Table 5 foods-13-00462-t005:** Trace element bioaccessibility (mg/Kg dry weight) in turnip tops with casein (mean ± standard deviation).

	Fe (mg/Kg)	Mn (mg/Kg)	Ni (mg/Kg)	Se (µg/Kg)	Zn (mg/Kg)
Turnip tops	28.4 ± 6.2 ^a^	21.9 ± 3.9 ^a^	1.05 ± 0.07 ^b^	9.2 ± 2.4 ^a^	16.7 ± 0.7 ^a^
Turnip tops + 5% casein	24.0 ± 2.7 ^a^	22.6 ± 2.3 ^a^	0.52 ± 0.09 ^a^	50.8 ± 4.0 ^b^	19.3 ± 1.5 ^a^
Turnip tops + 15% casein	26.3 ± 6.9 ^a^	24.1 ± 1.2 ^a^	0.53 ± 0.14 ^a^	50.8 ± 2.3 ^b^	24.3 ± 5.3 ^a,b^
Turnip tops + 25% casein	34.3 ± 3.6 ^a^	27.2 ± 0.8 ^a^	0.59 ± 0.16 ^a^	49.5 ± 0.5 ^b^	32.4 ± 0.1 ^b^

Within each column, means with different lowercase letters are significantly different at *p* < 0.05 according to the analysis of variance (ANOVA).

**Table 6 foods-13-00462-t006:** Trace element bioaccessibility (mg/Kg dry weight) in turnip tops with lactalbumin (mean ± standard deviation).

	Fe (mg/Kg)	Mn (mg/Kg)	Ni (mg/Kg)	Se (µg/Kg)	Zn (mg/Kg)
Turnip tops	28.4 ± 6.2 ^a^	21.9 ± 3.9 ^a^	1.05 ± 0.07 ^b^	9.2 ± 2.4 ^a^	16.7 ± 0.7 ^a^
Turnip tops + 5% lactalbumin	27.4 ± 4.6 ^a^	29.7 ± 0.5 ^a^	0.55 ± 0.06 ^a^	45.8 ± 2.4 ^b^	21.3 ± 2.1 ^a^
Turnip tops + 15% lactalbumin	26.6 ± 0.1 ^a^	25.1 ± 1.1 ^a^	0.59 ± 0.02 ^a^	46.2 ± 5.4 ^b^	23.6 ± 2.0 ^a^
Turnip tops + 25% lactalbumin	37.4 ± 8.6 ^a^	26.8 ± 2.2 ^a^	0.67 ± 0.01 ^a^	47.6 ± 3.6 ^b^	27.5 ± 7.1 ^a^

Within each column, means with different lowercase letters are significantly different at *p* < 0.05 according to the analysis of variance (ANOVA).

**Table 7 foods-13-00462-t007:** Trace element bioaccessibility (mg/Kg dry weight) in turnip tops with soy protein (mean ± standard deviation).

	Fe (mg/Kg)	Mn (mg/Kg)	Ni (mg/Kg)	Se (µg/Kg)	Zn (mg/Kg)
Turnip tops	28.4 ± 6.2 ^a^	21.9 ± 3.9 ^a^	1.05 ± 0.07 ^a^	9.2 ± 2.4 ^a^	16.7 ± 0.7 ^a^
Turnip tops + 5% soy protein	81.8 ± 3.0 ^b^	38.5 ± 4.5 ^b^	1.12 ± 0.51 ^a,b^	28.7 ± 2.4 ^b^	25.7 ± 4.2 ^a,b^
Turnip tops + 15% soy protein	75.6 ± 7.3 ^b^	30.2 ± 2.5 ^a,b^	0.88 ± 0.16 ^a^	52.7 ± 2.4 ^c^	34.4 ± 6.6 ^b^
Turnip tops + 25% soy protein	82.6 ± 2.1 ^b^	32.1 ± 3.0 ^a,b^	1.20 ± 0.09 ^b^	51.8 ± 1.8 ^c^	46.3 ± 1.6 ^c^

Within each column, means with different lowercase letters are significantly different at *p* < 0.05 according to the analysis of variance (ANOVA).

## Data Availability

The data presented in this study are available in article.

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
