# Peer review of "Influence of Dietary Fibre and Protein Fractions on the Trace Element Bioaccessibility of Turnip Tops (Brassica rapa) Growing under Mediterranean Conditions"

_foods, 2024, doi:10.3390/foods13030462_

Round 1
Reviewer 1 Report
Comments and Suggestions for Authors
Manuscript foods-2808326
In this study, the authors reported a study concerning the effect of several dietary fibre and protein fractions on the trace element bioaccessibility of turnip tops (B. rapa subsp. Rapa). The bioaccessibility protocol was based on the study of Cámara et al. 2005, which simulates in vitro the oral, gastric, and intestinal stages of human digestion. The manuscript is well-written, and the English language is adequate.
I have few suggestions for authors consideration:
1) Line 109: Please consider including a comment concerning these percentages. Is it common to find them in food mixtures (lines 71-73)?
2) Item 2.3. Bioaccessibility assay: Please include the enzymes and their specifications (P/N or activity). In addition, please confirm the information concerning a-amylase and pepsin solutions (lines 111-115).
3) Lines 152-155: Please consider including the range used for analytical curves.
4) Lines 201-204: It can be noticed that some protocols, such as INFOGEST, use pH near 7.0 for the intestinal stage. Please consider including few comments.
Author Response
First of all, I would like to thank the reviewer for his kind comments and the time taken to review the manuscript.
I have few suggestions for authors consideration:
1) Line 109: Please consider including a comment concerning these percentages. Is it common to find them in food mixtures (lines 71-73)?
Thank you for your valuable suggestion. I included some information about this topic. Please, see lines 198 – 199; 275 – 277; 318 – 319; 414 – 415.
2) Item 2.3. Bioaccessibility assay: Please include the enzymes and their specifications (P/N or activity). In addition, please confirm the information concerning a-amylase and pepsin solutions (lines 111-115).
Done. Please, see lines 91 – 93 and 117 – 121.
3) Lines 152-155: Please consider including the range used for analytical curves.
Done. Please, see lines 164 – 165. Thank you.
4) Lines 201-204: It can be noticed that some protocols, such as INFOGEST, use pH near 7.0 for the intestinal stage. Please consider including few comments.
Done. Please, see lines 219 - 221

Reviewer 2 Report
Comments and Suggestions for Authors
General comment:
The present work (foods-2808326) addresses a very relevant topic, namely the impact of different antinutrients on the bioaccessibility of minerals in turnips. The work is well-designed. However, the experimental procedure is not well-defined, leading me to consider the possibility of fundamental flaws that may have compromised the obtained results. The author needs to describe in great detail how the bioaccessibility assay was conducted, specifying, among other things, the concentrations and activities of the enzymes used, as well as the volumes at each stage of the procedure. Furthermore, it is unclear how the bioaccessible fraction was mineralized. What volume was effectively utilized? This information, and others detailed below, is crucial for a better understanding of the presented results since some of these results (e.g., pectin) are very controversial. That is, the "negative" effect of pectin on mineral bioaccessibility due to its proven ability to bind to these elements, preventing their absorption, is well reported in the available scientific literature. However, the authors present results contrary to this evidence, showing that the addition of pectin increases, for example, the bioaccessibility of Fe. Well, I find this highly improbable. Therefore, at this stage, I consider the manuscript not ready for acceptance, and more information needs to be provided by the author to clarify these fundamental issues.
Specific Comments:
(L.41) Additional works need to be included to substantiate this statement. Furthermore, reference 10 constitutes a self-citation. The author should broaden their collection of available scientific literature on the topic.
(L.105) Which vessels were used for this assay? 50 ml centrifuge tubes? If yes, specify.
(L.111) Add the concentration/activity of the amylase solution used.
(L.114) This doesn't seem correct. Did the authors weigh 0.083 g of a pepsin solution? If it was a solution, shouldn't a volume be added? If a powder was used, the sentence should be rewritten. Additionally, why this amount? Typically, proper enzyme activity needs to be achieved. Was this the case?
(L.117) Refer to my previous comment. Why were these amounts considered? Were they used to achieve adequate enzyme activity? If yes, specify.
(L.120) At this point, it is challenging for me to understand the final volume of the obtained solution. Was the volume adjusted to a specific value before the transfer step? Was the final volume the same for every replicate and condition?
(L.125) Explain exactly how the blanks were prepared. Were all reagents and fiber/protein fractions without turnip tops used?
(L.129) The sentence is confusing. Clearly rewrite that this procedure is for the determination of the bioaccessible fraction of trace elements. Remove "the determination of trace elements in turnip tops samples."
(L.130) The authors need to specify the volume of the bioaccessible fraction they used to conduct the sample mineralization. Add this information.
(L.156) Information on the LOD/LOQ of the methods used needs to be added. Provide this information.
(L.171) None of these papers determined the trace element content in turnip tops. How can the authors compare those results with these? In all works, samples from Brassica species were studied, but none from turnip tops. Am I correct? Furthermore, all four citations are from the authors of this paper, resulting in a poor comparison and discussion. Authors should expand their portfolio of available scientific papers on this subject.
(L.175-176) Poor English writing. Rewrite this sentence.
(L.182) This contradicts the vast body of literature that affirms and proves that pectin is an antinutrient with a high binding capacity for minerals, and its presence reduces the bioaccessibility of minerals. Even under the conditions of the bioaccessibility assay, several papers have proven that pectin binds to minerals and reduces their bioaccessible fraction. How is it possible that pectin increases, for example, Fe and Zn bioaccessibility?
(Table) Pectin. Correct"
Comments on the Quality of English LanguageThe English quality is subpar and necessitates substantial enhancements. The manuscript contains poorly constructed sentences, making it challenging to read. I highly recommend utilizing a language editing service
Author Response
(L.41) Additional works need to be included to substantiate this statement. Furthermore, reference 10 constitutes a self-citation. The author should broaden their collection of available scientific literature on the topic.
Thank you for your suggestion. I agree with the reviewer. I talked about different studies but I only showed a self- citation. My apologies for this mistake. I have included two more references.
Mataix-Verdú, J., & Llopis-González, J. (2015). Minerales. Nutrición y alimentación humana (second ed.). Madrid (Spain): Ergón265–301.
Kamchan, A., Puwastien, P., Sirichakwal, P. P., & Kongkachuichai, R. (2004). In vitro calcium bioavailability of vegetables, legumes and seeds. Journal of Food Composition and Analysis, 17(3–4), 311–320. https://doi.org/10.1016/j.jfca.2004.03.002.
(L.105) Which vessels were used for this assay? 50 ml centrifuge tubes? If yes, specify.
Done. Please, see line 112
(L.111) Add the concentration/activity of the amylase solution used.
Done. Please, see lines 91 – 93 and 117
(L.114) This doesn't seem correct. Did the authors weigh 0.083 g of a pepsin solution? If it was a solution, shouldn't a volume be added? If a powder was used, the sentence should be rewritten. Additionally, why this amount? Typically, proper enzyme activity needs to be achieved. Was this the case?
I have specified this point in line 120. The pepsin solution was prepared in 0.1 N HCL.
The amount of pepsin used corresponds to a protocol standardised by different research groups. We have been working in the topic of trace element bioaccessibility for 25 years:
Sahuquillo et al. (2003). Nahrung, 47(6), 438–441
Periago et al. (1996). LWT – Food Science and Technology 5 – 6, 481-488
Ramírez – Ojeda et al. (2017) European Food Research Technology 243, 639–650
Toreti Theodoropoulos et al. (2018) Food Research International 108 68–7
Martínez – Castro et al. (2023). Journal of Trace Elements in Medicine and Biology 78 127181
And many more…
The pepsin amount is related to the weight of food digested (2 g in this study). In addition, I have indicated the enzyme activity in line 120
We have seen that better results are obtained following this methodology than other standardised protocols
(L.117) Refer to my previous comment. Why were these amounts considered? Were they used to achieve adequate enzyme activity? If yes, specify.
I have already answered this question in the previous suggestion.
(L.120) At this point, it is challenging for me to understand the final volume of the obtained solution. Was the volume adjusted to a specific value before the transfer step? Was the final volume the same for every replicate and condition?
The final digested volume before centrifugation is irrelevant. It should be considered that this mixture is going to be centrifuged and then we are going to collect the supernatant. Finally, as indicated in lines 138 - 146, this supernatant will be dried on a hot plate and incinerated. It is the blanched ashes which are dissolved in 10 mL. This is the volume that was used to perform the calculations (along with the 2 g of sample).
However, at the reviewer's suggestion, I have indicated this point in the line 128
(L.125) Explain exactly how the blanks were prepared. Were all reagents and fiber/protein fractions without turnip tops used?
This aspect was already clarified in the lines 113 - 116. But in case it was not clear, I have provided this information again on the lines 135 - 136. Thank you for your suggestion.
(L.129) The sentence is confusing. Clearly rewrite that this procedure is for the determination of the bioaccessible fraction of trace elements. Remove "the determination of trace elements in turnip tops samples."
I agree with you. Thank you for your suggestion
(L.130) The authors need to specify the volume of the bioaccessible fraction they used to conduct the sample mineralization. Add this information.
Same consideration as before. This information is irrelevant. The supernatant is going to be dried on a hot plate and incinerated. It is the blanched ashes that are dissolved in 10 mL, which is the volume that was used to perform the calculations (along with the 2 g of sample).
(L.156) Information on the LOD/LOQ of the methods used needs to be added. Provide this information.
I provided this information in lines 169 - 171. Thank you for your suggestion
(L.171) None of these papers determined the trace element content in turnip tops. How can the authors compare those results with these? In all works, samples from Brassica species were studied, but none from turnip tops. Am I correct? Furthermore, all four citations are from the authors of this paper, resulting in a poor comparison and discussion. Authors should expand their portfolio of available scientific papers on this subject.
I'm afraid that in this case, the reviewer is wrong. In these four previous works turnip tops (Brassica rapa) were analyzed. I suggest you check it out.
On the other hand, my research group has been studying the nutritional benefits of cruciferous vegetables (turnip tops among them) for a long time. As I indicated in lines XX “in recent years, a programme for the adaptation of varieties of B. rapa subsp. rapa to the climatic and environmental conditions of southern Spain has been developed”.
Although there is data on mineral and trace element concentration in other cruciferous (mainly Brassica oleracea), as far as I know, in the case of turnip tops we only have those previously published by us.
(L.175-176) Poor English writing. Rewrite this sentence.
I regret this comment from the reviewer. As it appears in the acknowledgments section, the English of the manuscript was reviewed by a native speaker.
I have sent it to her again to review. As can be seen (except for the abstract that she did not review), the English of the manuscript is correct.
(L.182) This contradicts the vast body of literature that affirms and proves that pectin is an antinutrient with a high binding capacity for minerals, and its presence reduces the bioaccessibility of minerals. Even under the conditions of the bioaccessibility assay, several papers have proven that pectin binds to minerals and reduces their bioaccessible fraction. How is it possible that pectin increases, for example, Fe and Zn bioaccessibility?
I disagree with the reviewer's comment. As I indicate in different parts of my manuscript (supported by references), the role of soluble fibre in impairing trace element bioaccessibility has been unclear for some time.
In the case of pectin, I support this statement with previous work:
Peixoto, R.A.; Devesa, V.; Vélez, D.; Cervera, M.L. ; Cadore, S. Study of the factors influencing the bioaccessibility of 10 elements from chocolate drink powder. J. Food Compos. Anal. 2016, 48, 41–47. http://dx.doi.org/10.1016/j.jfca.2016.02.002.
Miyada, T.; Nakajima, A.; Ebihara, K. Iron bound to pectin is utilised by rats. Br. J. Nutr. 2011, 106, 73 - 78. http://dx.doi.org/10.1017/S0007114510005842.
Table) Pectin. Correct"
Done. Thank you for your suggestion.
Comments on the Quality of English Language
The English quality is subpar and necessitates substantial enhancements. The manuscript contains poorly constructed sentences, making it challenging to read. I highly recommend utilizing a language editing service.
I have already responded to this comment before.
